# Quality of medicines for Cardio-Vascular Diseases (CVDs) in the Ethiopian border with Kenya: The case of enalapril maleate and furosemide tablet quality in Borena and Gedeo zones

Esayas Tesfaye[1,2], Fisseha Woldegiorgis[3], Tadele Eticha[1], Ayenew Ashenef[1,4]*

1 Department of Pharmaceutical Chemistry and Pharmacognosy, School of Pharmacy, College of Health Sciences, Addis Ababa University, Addis Ababa, Ethiopia, 2 Department of Pharmacy, College of Health Science, Dilla University, Dilla, Ethiopia, 3 Humanwell Pharmaceutical Ethiopia PLC, North Shoa, Tulefa, Ethiopia, 4 Center for Innovative Drug Development and Therapeutic Trials for Africa (CDT-Africa), College of Health Sciences, Addis Ababa University, Addis Ababa, Ethiopia

* ayenew.ashenef@aau.edu.et

**Data Availability Statement:** The datasets used in this publication are included in the paper and the supplementary files.

## Abstract

Hypertension (HTN), a cardiovascular disease (CV), is a major public health challenge. Therefore, the quality of drugs used to treat it has become a major concern. Enalapril and furosemide are among the drugs prescribed to manage hypertension. Hence, this study is aimed at evaluating the quality of different brands of enalapril and furosemide tablets available in the Gedeo and Borena zones, southern Ethiopia. Thirteen generic brands of enalapril maleate (5 mg) and furosemide (40 mg) tablets were evaluated for visual defects, *in vitro* dissolution test, weight variations, friability, hardness, and disintegration tests. The analysis of active pharmaceutical ingredient (API) content was performed by high performance liquid chromatography (HPLC). Out of 55 samples, 7 (12.73%) failed to comply with the criteria for visual inspection; otherwise, all samples passed the identification test. Except for one brand of each of enalapril maleate and furosemide, all passed the dissolution test. The assay value showed that all enalapril maleate samples were within the limits of United States Pharmacopoeia (USP), 2020. Additionally, except for two batches, all furosemide samples were within the USP and British Pharmacopoeia (BP), 2020 standards. Out of 55 samples, 8% (2/25) and 6.67% (2/30) of enalapril maleate and furosemide tablets failed the uniformity of dosage units test per the USP-2020, respectively. All samples passed the disintegration test, and selected furosemide samples passed the microbial limit tests. However, 36% (9/25) and 20% (6/30) of enalapril maleate and furosemide samples failed to pass hardness test. Generally, from the total samples, 50.91% (28/55) were substandard (did not meet the specifications failing any one or more parameters assessed). The studied drugs circulating in the market did not meet some of the needed quality specifications. This could have brought a risk of reduced efficacy due to the distribution of poor quality medicines in the area.

**Funding:** The study was sponsored by the Addis Ababa University Graduate research support program to ET and the Ministry of Innovation and Technology, government of Ethiopia, by the project scheme National Problem Solving award to AA.

**Competing interests:** The authors declare that they did not have any conflicts of interest to declare in relation to this study and publication. The funders did not have any role in the study design, data interpretation or publication.

## Background

Cardiovascular disease is the term used to describe diseases affecting the heart and circulatory system, including stroke and raised blood pressure (hypertension) [1]. In 2017, the American College of Cardiology/American Heart Association Task Force on Clinical Practice Guidelines redefined hypertension in adults as systolic blood pressure $\geq$ 130 mmHg and/or diastolic blood pressure $\geq$ 80 mmHg [2]. Most patients with hypertension (HTN) require drug treatment to achieve sustained reduction of blood pressure. Loop diuretics or high-ceiling, $\beta$-blockers, calcium antagonists, angiotensin converting enzyme inhibitors (ACE-Is) and angiotensin II receptor blockers (ARBs) can adequately lower blood pressure and reduce the risk of cardiovascular (CV) death and morbidity [3]. Thus, these drugs are all recommended for the initiation and maintenance of blood pressure control, either as monotherapy or in combination for an additive or synergistic effect.

Cross border trade between Ethiopia and Kenya is an important feature that is existing for many years. Informal cross-border trade between Ethiopia and Kenya is substantial and vital for both countries [4]. Our study area Moyale and Mendera are the two main trading hubs across the border between Kenya and Ethiopia. Ethiopia's main exports include livestock, livestock products and cereals. On the other hand, Kenya's exports to Ethiopia are manufactured products including processed foods. Similarly cross-border trade in medicine in the borderlands of Ethiopia and Kenya exists but it is not, as such, recorded adequately [5]. Hence we are assessing the quality of selected medicines that may enter to Ethiopia through informal means without the proper national medicine authority (NMA) regulatory scrutiny on medicine importation process.

This study was focused on enalapril maleate (angiotensin converting enzyme inhibitor) and furosemide (diuretics), which are used in the management of HTN [6, 7].

Enalapril maleate is used to prevent, treat or improve the symptoms of hypertension. It is also used in the management of symptomatic heart failure or asymptomatic left ventricular dysfunction, coronary artery disease, and certain chronic kidney diseases. It had a favorable efficacy and tolerability profile [8]. It is a prodrug that is bioactivated by hydrolysis of the ethyl ester to enalaprilat, which is the active angiotensin converting enzyme inhibitor. Inhibition of angiotensin converting enzyme (ACE) results in decreased plasma angiotensin II levels, which leads to decreased vasopressor activity and aldosterone secretion [9]. It is chemically described as (S)-1-[N-[1-(ethoxycarbonyl)-3-phenylpropyl]-L-alanyl]-L-proline, (Z)-2 butenedioate salt (1:1) with the empirical formula $C_{20}OH_{28}N_2O_5 \bullet C_4H_4O_4$ [10]. The structural formula of enalapril maleate is given in Fig 1A,. It is a white to off-white, crystalline powder with a molecular weight of 492.53 [11]. The WHO only lists strengths of 2.5 and 5.0 mg in their list of essential drugs [12].

Furosemide is the most commonly used loop diuretic (furosemide, bumetanide, torsemide and ethacrynic acid) [9]. These drugs have the highest efficacy in mobilizing $Na^+$ and $Cl^-$ from the body. It inhibits the cotransport of $Na^+/K^+/2Cl^-$ in the luminal membrane in the ascending limb of the loop of Henle [13]. Chemically, it is 5 (aminosulfonyl)-4-chloro-2-[(2-furanyl-methyl)-amino] benzoic acid. It is a white to slightly yellow, odorless, crystalline powder with a molecular weight of 330.77 [14]. Its empirical formula is $C_{12}H_{11}ClN_2O_5S$, and the structural formula is given in Fig 1B, [12, 13]. The bioavailability of furosemide from oral dosage forms is highly variable due to its poor solubility. Thus the difference in formulation aspects from brand to brand as a result of the manufacturer's technicalities will make their pharmacokinetic profile prone to high variations [15].

Several studies have been carried out to assess the quality of enalapril maleate and furosemide tablets in some countries across the globe. In a study conducted in India, most of the

**Fig 1.** Structural formulae of (A) enalapril maleate and (B) furosemide.

enalapril maleate brands passed the test parameters except for variation in hardness, disintegration time and dissolution profile during the test procedure [16]. Another study conducted in Brazil and Guatemala revealed that all samples of enalapril tablets showed satisfactory test results per pharmacopoeial specifications. In Turkey, study samples complied with the pharmacopoeia standards except hardness and friability properties [17–19]. A study conducted in Brazil and Argentina also revealed that all brands of furosemide tablets passed the dissolution test profiles [20–22].

In a study conducted in Nigeria, except for a few samples that failed in hardness, most of the furosemide tablet batches tested met with label claims in terms of the content of furosemide and dissolution profile [23]. A similar study in 10 sub-Saharan countries revealed that the prevalence of poor-quality drugs differed significantly between drugs, including a 12.5% substandard and falsified (SF) prevalence for furosemide. The proportion of poor-quality drugs exceeded 20% in other African countries [24]. However, a similar study in Libya revealed that all brands were in compliance with the specifications [7].

A study conducted in the Ethiopia capital city Addis Ababa showed that the majority of furosemide brands passed quality attributes tested but failed in hardness and *in vitro* drug dissolution profiles [25]. However, a study performed in Bahir Dar city, Northwest Ethiopia, revealed that all furosemide brands met all quality specifications per the pharmacopoeial specifications [26].

The aim of this study was to evaluate the quality of enalapril and furosemide samples collected from southern Ethiopia, specifically from the Gedeo and Borena zones. These areas were targeted due to existence of cross-border illegal trade in medicines at the Ethiopian-Kenyan border.

## Materials and methods

### Ethics approval and consent to participate

Ethical approval was obtained from the School of Pharmacy, Addis Ababa University Ethical Review Board (ERB) referenced as ERB/SOP/361/13/2021. Consents and other ethical aspects of the work had been guided on the current principles of research.

### Materials

Enalapril maleate and furosemide primary standards available at Humanwell Pharmaceutical Company PLC were used. Methanol for HPLC was purchased from Merck Life Science Private Limited, Germany. Sodium hydroxide pellets (or pearls) 97% Extra Pure were purchased from Loba Chemie Pvt. Ltd. Mumbai, India. Tetrahydrofuran, glacial acetic acid and phosphoric

acid (85%) were purchased from sd fine chEM, Loba Chemie Pvt. Ltd., Mumbai and Delhi, India, respectively. Monobasic potassium and sodium phosphate were purchased from Loba Chemie Pvt. Ltd, Mumbai, India. Acetonitrile was purchased from Merck Life Science Pvt Ltd, Germany. Soybean-casein digest agar (SCDA), soybean-casein digest broth (SCDB), Sabour-aud dextrose agar (SDA), MacConkey agar (MCA), and MacConkey broth (MCB) were obtained from Beijing, China.

## Samples of medicines

All 55 samples of medicines (that includes enalapril maleate (25/55) and furosemide (30/55)) were purchased for quality testing from retail pharmacies and hospitals in Gedeo and Borena Zones, southern Ethiopia. The sample collection sites are shown on a map as depicted in Fig 2, (constructed using the QGIS software).

Both medicines were within their expiration dates before the test. The details of the two sampled medicines were described in S1 and S2 Files. Samples were collected from 39 drug

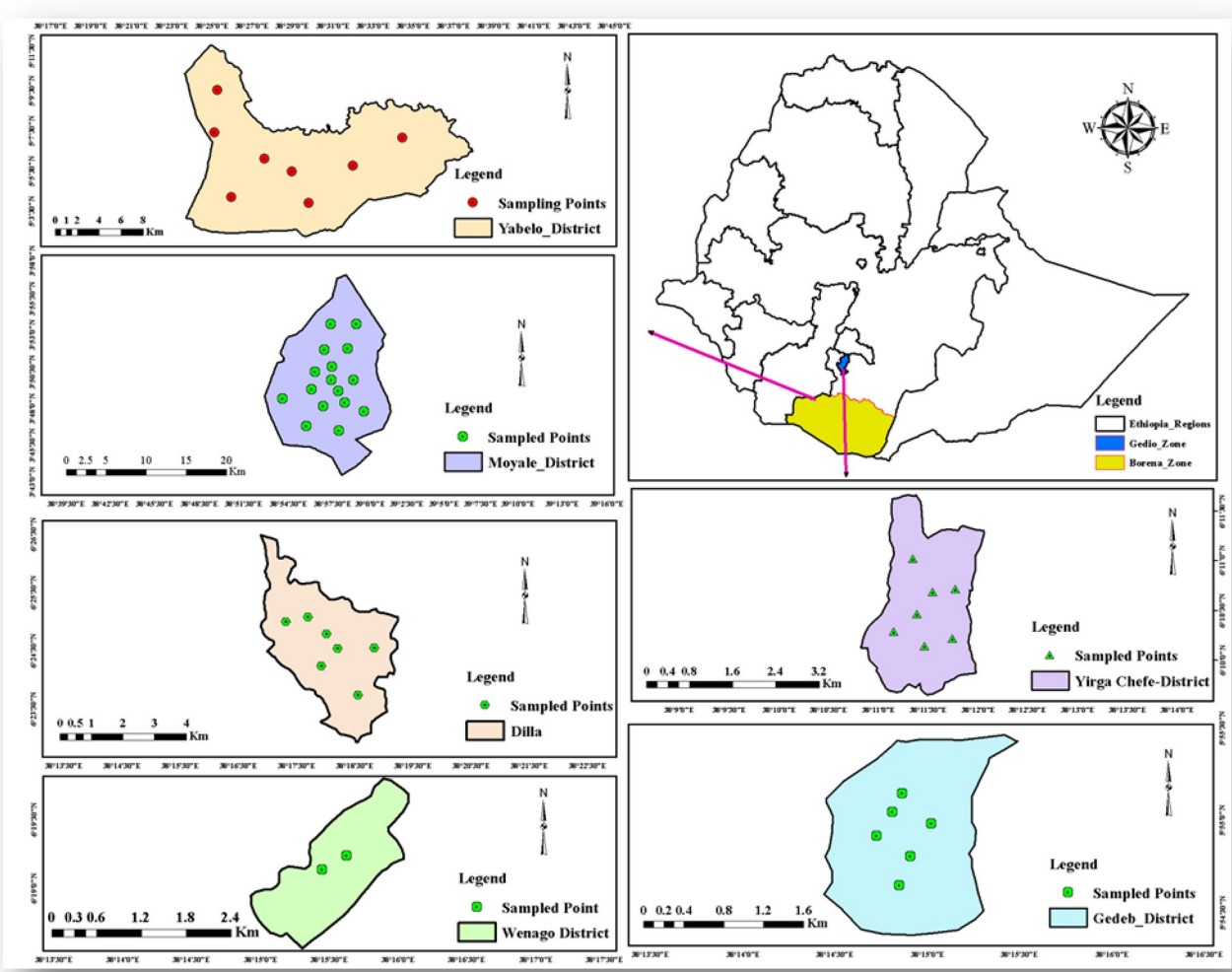

**Fig 2. Map of the study area (Gedeo and Borena zones) (Base line map from https://www.qgis.org/en/site/) constructed using the QGIS software under CC 4.0.**

outlets (S3 File), of which 86.8% of samples were collected from private facilities, whereas the remaining 13.2% were collected from government facilities. The map depicting the sampling sites is given in Fig 2, derived from their GPS coordinates. During the sample collection period, six and seven brands of enalapril maleate and furosemide tablets (S4 File) were found respectively in the market of the geographical area covered in this investigation.

### Inclusion and exclusion criteria

Samples were collected based on inclusion and exclusion criteria. The study was ethically approved by the Addis Ababa University School of Pharmacy Ethical Review Committee/board as ERB/SOP/361/13/2021 dated September 29, 2021. In addition, a letter of cooperation was written from the Ethiopian Food and Drug Authority Moyale entry and exit port branch office to support the study in that locality. The inclusion criteria were to collect all types of tablet dosage forms of enalapril maleate and furosemide available in a given retail outlet selected for this study. The exclusion criterion was to avoid collection of the same product with a similar batch number at the same sample collection site.

### *In vitro* quality evaluation of enalapril maleate and furosemide

The parameters assessed include visual inspection by the Joint WHO/USP/FIP check list [27]. Identity tests were performed based on USP and BP methods [28, 29]. Assay was determined according to the pharmacopoeial methods [30]. Uniformity of dosage units was assessed based on USP 2020 method. Test for hardness, test for friability, disintegration and dissolution tests were also assessed in accordance with the pharmacopoeial methods. Microbial methods i.e. total aerobic microbial count, total yeast and mold count and *E. coli* assessments were also based by pharmacopoeial methods [31].

The laboratory works were performed in the National Medicine regulatory agency (NMRA) current good manufacturing practice (CGMP) certified pharmaceutical industry quality control laboratory. Experiments were done per their standard operating procedures (sops), that were in place to assure adherence to all quality assurance procedures. Thus, the reliability of the data generated is guaranteed to current standards of the industry.

## Results

### Physical characteristics, packaging and labeling information

According to visual inspection of the WHO, EU and USP checklists, the majority of enalapril maleate and furosemide tablets sampled were well blistered and packed properly. The packages were appropriately labeled. However, there were seven (12.72%) samples: three from enalapril maleate and four from furosemide with poor packaging. A leaflet or package insert was absent in them. Few furosemide samples were cracked, eroded and caked. From a total of 55 tablets, 12% (3/25) of enalapril maleate and 13.33% (4/30) of furosemide tablets had poor physical characteristics and appearances, respectively. Tables 1 and 2 explain failed samples on packaging and physical characteristics. Detailed descriptions and images of failed visual tests were fully shown on the S1 Fig in the visual defects section.

### Identification

**Identification test results of enalapril by HPLC.** The identity of the tested products was confirmed by comparing their retention times with that of the enalapril reference standard. All enalapril samples analyzed displayed retention times corresponding with those of the respective reference standards. Fig 3, depicts one representative enalapril sample code, EM-10,

**Table 1. Physical characteristics of the tablets.**

| Sample (Brand) | Uniformity of colour | Uniformity of size | Uniformity of shape | Surface spot or contamination | Breaks, and cracks |
|---|---|---|---|---|---|
| Enalapril (Korandil) | ✓ | ✓ | ✓ | X | X |
| Enalapril (Enali-SSP) | ✓ | ✓ | ✓ | ✓ | X |
| Furosemide (Fusix) | ✓ | ✓ | ✓ | ✓ | X |
| Furosemide (Fruz)* | ✓ | ✓ | X | X | X |
| Furosemide (Frusemide) | ✓ | ✓ | ✓ | X | ✓ |

'X' = Failed sample (there are non-uniform shapes, surface spots or contaminations observed and breaks or cracks observed), '✓' = Passed sample (color, shape, size uniform; absence of surface spot or contamination and breaks as well as cracks

collected from Moyale, which contained the correct API compared to its reference standard enalapril maleate chromatogram.

**Identification test results of furosemide by UV.** From the total of 30 samples, 20 samples were manufactured in compliance with the USP specification. Thus, an identity test was performed according to USP-43, 2020. The UV absorbance maxima and minima results for the furosemide reference standard were 227.00 nm and 332.42 nm, respectively. Similarly, every sample had exhibited maxima and minima UV absorbance wavelength very close to the wavelength of the furosemide standard. Fig 4, depicts the identification test of the standard and a representative sample coded FW-01 of furosemide collected from the Wenago site.

Similarly, the remaining 10 samples of furosemide were manufactured per BP specification; therefore, identification tests were performed according to BP 2020. By scanning in the range 220 to 320 nm of the final solution obtained in the assay, the samples exhibit two maxima, at 228 nm and 271 nm. In these tests, the samples exhibit absorption maxima from 226.609 to 228.565 nm and absorption minima from 269.565 to 269.840 nm. There were few wavelength deviations observed in both the absorption maxima and minima. The details obtained in the identification tests for furosemide are included in the S5 and S6 Files.

## Assay

**Assay result of enalapril maleate.** The assay values for enalapril drug products ranged from 91.35% to 104.53%. This showed that all batches comply with the USP specification limit (90–110%). All assay values are given in S7 File.

*System suitability test results for the enalapril maleate assay*. Five replicates of 50 μL enalapril maleate USP reference standard solutions were injected into the HPLC system, and the chromatograms were recorded to evaluate the system suitability parameters as the tailing factor (Not more than (NMT) 1.5), theoretical plate number (Not less than (NLT) 300 for enalapril), and % RSD (NMT 2.0). Accordingly, the system suitability test criterial were met before the sample analyses.

**Table 2. Packing and labeling information for the different brands.**

| Product code | Medicine strength (mg/tablet) | Dosage Statement | Batch/Lot no. | Storage condition | Manufacturing date | Expiry date |
|---|---|---|---|---|---|---|
| Enalapril (Korandil) | ✓ | ✓ | ✓ | ✓ | 11/2020 | 11/2023 |
| Enalapril (Enali-SSP) | ✓ | ✓ | ✓ | ✓ | 11/2020 | 10/2022 |
| Furosemide (Fusix) | ✓ | ✓ | ✓ | ✓ | 06/21 | 06/24 |
| Furosemide (Fruz) | ✓ | ✓ | ✓ | ✓ | 04/2021 | 03/2024 |
| Furosemide (Frusemide) | ✓ | ✓ | ✓ | ✓ | 05/2021 | 04/2024 |
| Furosemide (Fruz) | ✓ | ✓ | ✓ | ✓ | 11/2020 | 11/2023 |

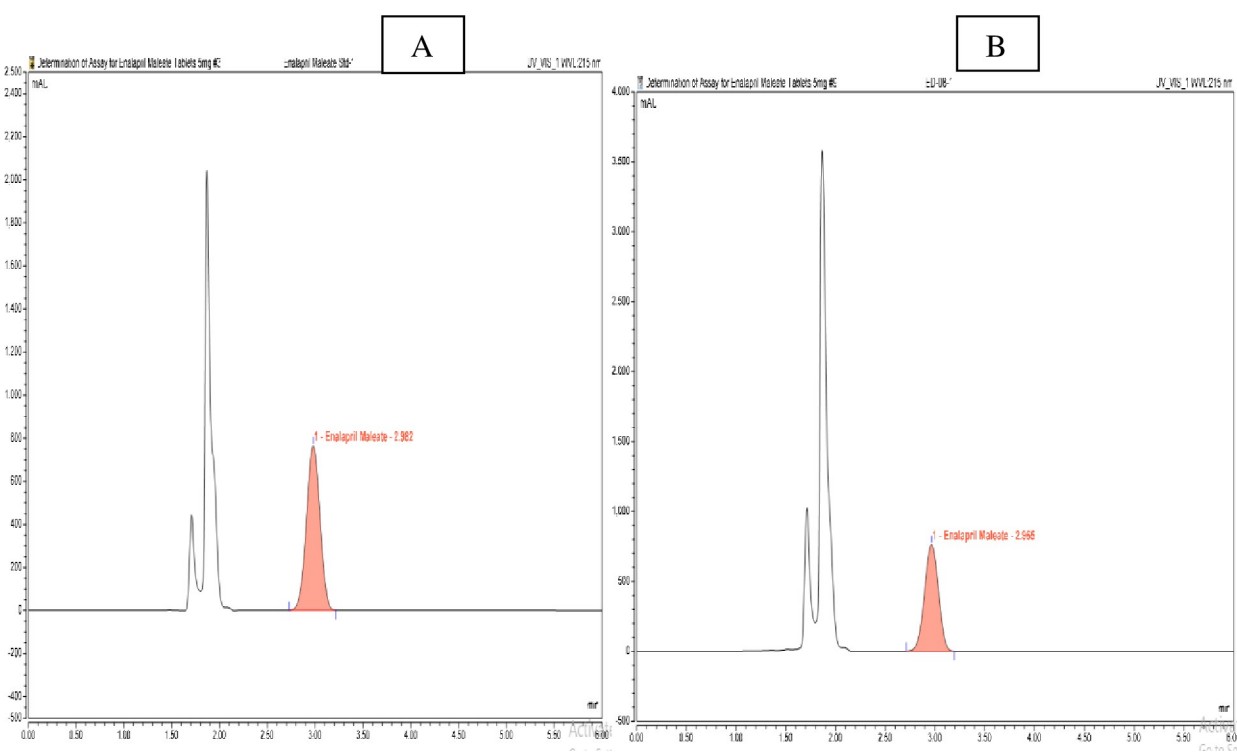

**Fig 3.** (A) Peak retention time of standard enalapril maleate and (B) sample EM-10.

*HPLC analysis of enalapril.* First, the calibration curve was prepared using 6 concentrations within the range of 0.08 mg/mL to 0.28 mg/mL enalapril. The internal standard was added to each sample, and the peak area ratio was plotted against the concentration (Fig 5). The calibration curve was found to be linear in the range with a regression factor $R^2 = 0.9998$.

**Assay result of furosemide.** *Assay results using BP and USP 2020.* From 30 samples, 10 samples were analyzed using the BP method, while for the rest 20 samples, the USP method was employed. The *absorbance value* of the resulting solution at the maximum wavelength 271 nm was measured. The minimum and maximum stated content (Assay) of furosemide samples were 92.33% and 101.11%, respectively. All except two had passed the BP specification limit (95–105%). The assay value performed using HPLC for furosemide drug products ranged from 91.68% to 101.22%. These values are in compliance with the USP specification. The detail values about assay results for each sample are shown on S8 and S9 Files.

## HPLC system suitability test result of the furosemide standard

Five replicates of 20 μL of furosemide USP reference standard solutions were injected into the HPLC system, and then chromatograms were recorded to evaluate the system suitability parameters, such as the tailing factor (NMT 1.5), theoretical plate number (NLT 2000 for furosemide), and % RSD (percent relative standard deviation) (NMT 2.0). According to the above parameters, the system suitability test data fulfilled the requirements.

## Calibration curve of the furosemide standard for the assay test

For the analysis of the drug, the calibration curve was prepared using 6 concentrations within the range of 0.4 mg/mL to 1.4 mg/mL furosemide. The concentrations of furosemide RS against the peak area ratio were plotted to obtain the calibration curves. The calibration curve

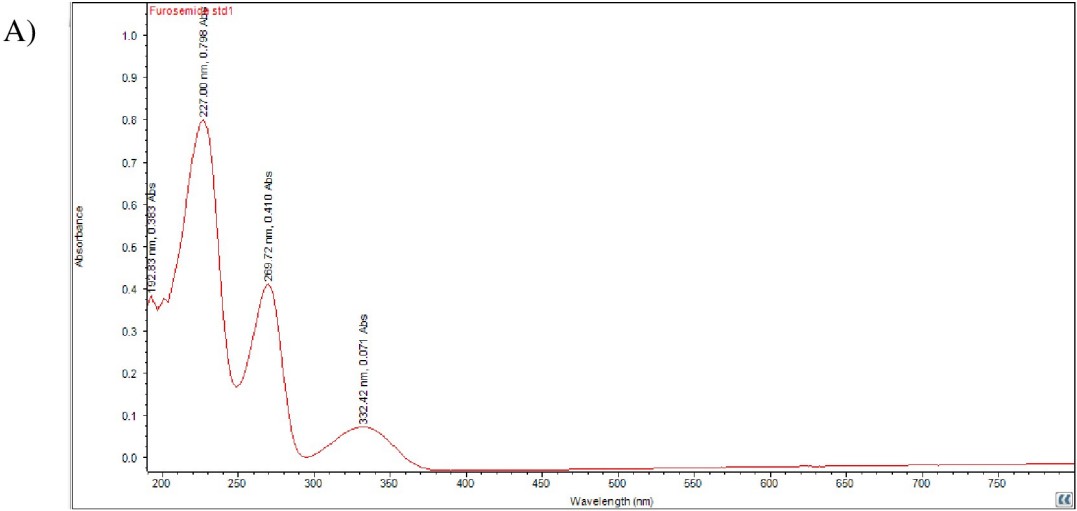

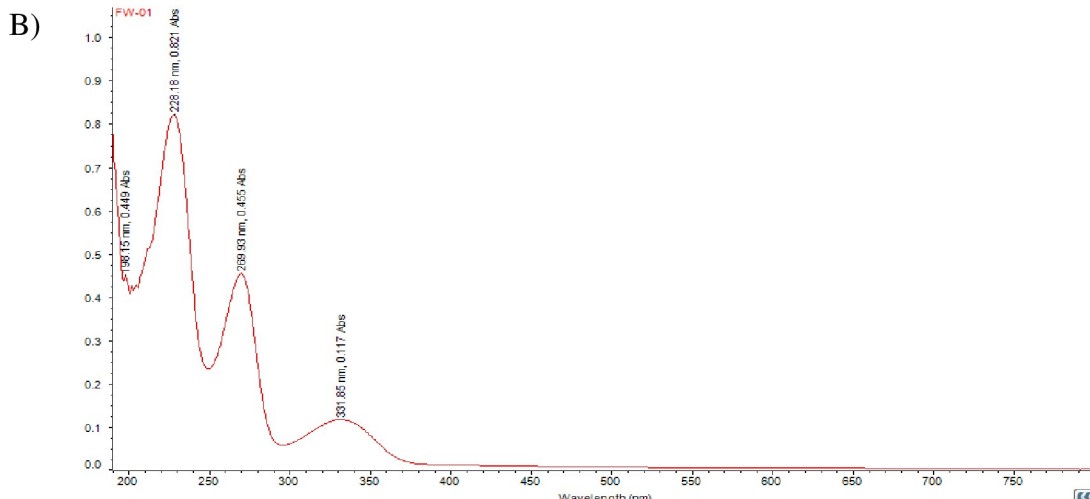

**Fig 4.** (A) Identification test result of furosemide standard and (B) FW-01 (USP-2020).

was found to be linear in the range with a regression factor $R^2$ = 0.9996. Fig 6, depicts the chromatogram of a standard sample and representative furosemide sample code FM-06 collected from Moyale city.

## Hardness test

This study revealed that from the total of 55 samples, 36% (9/25) and 20% (6/30) of enalapril maleate and furosemide samples respectively failed. These samples were from the Fusix brand. Table 3 lists details of the samples that failed the hardness test. The maximum and minimum mean hardness test results for enalapril maleate tablets were 104.7 ± 12.3 and 26.9 N ± 5.8, respectively. Analogously for the furosemide tablets, the values were 90.5 N ± 7.9 and 9.0 N ± 1.9, respectively.

## Disintegration time test

According to this study, all the samples passed the test. The maximum mean disintegration time test results for enalapril maleate and furosemide tablets were 7.33 ± 3.01 and 11.67 ± 2.25

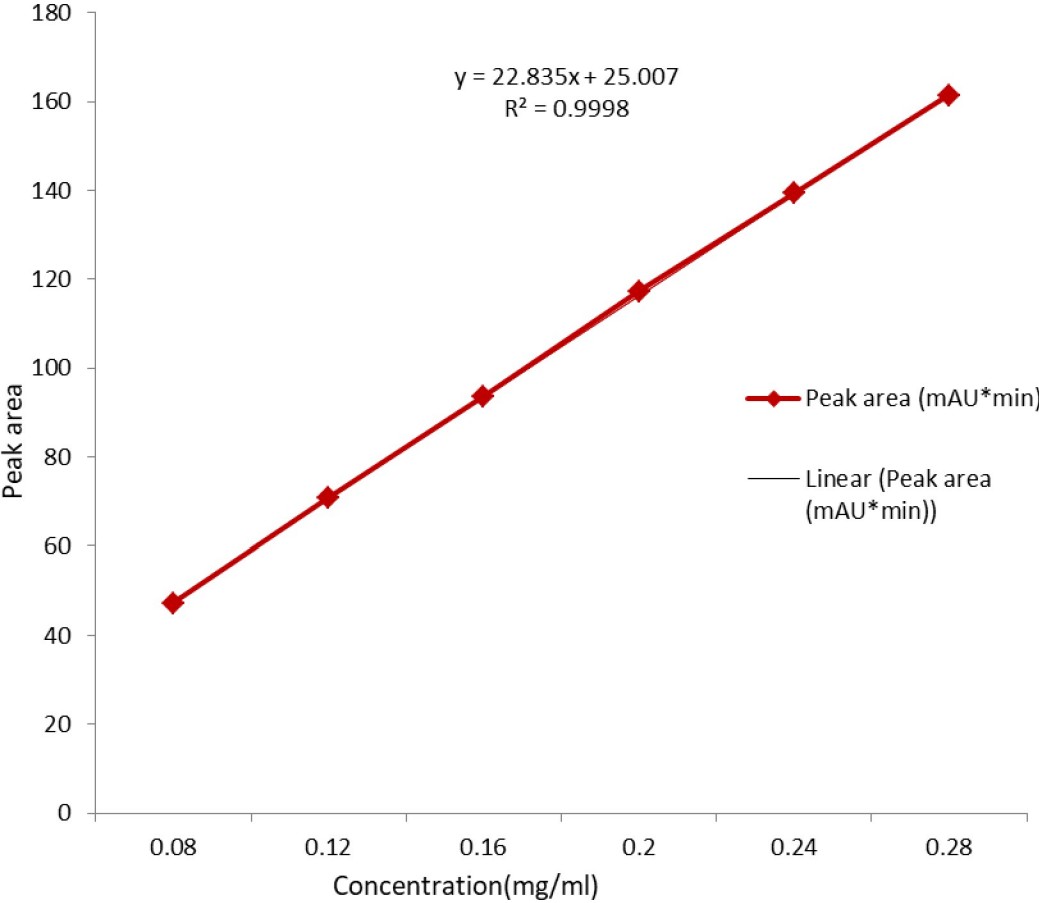

**Fig 5. The assay calibration curve of enalapril maleate RS (USP-2020).**

minutes, respectively. The maximum mean disintegration time of enalapril maleate was observed for the coated tablet. However, their minimum average disintegration time test results for enalapril maleate and furosemide tablets were $1.17 \pm 0.41$ minutes and $1.33 \pm 0.52$ minutes, each in order.

## Friability test

Most batches of the Fusix brand failed to comply with the USP specification for the friability test. Of the 55 samples, 12% (3/25) and 33.33% (10/30) of enalapril maleate and furosemide samples, respectively, failed the friability test. The percentage friability test results for enalapril maleate and furosemide tablets ranged from 0% to 5.07% and from 0% to 5.27%, in the mentioned order.

The brands Enali-SSP and Encardil-5 of enalapril maleate failed the friability test according to the USP specification. Similarly, these failed samples are not registered in the Ethiopian regulatory information system (ERIS) of the Ethiopian Food and Drug Authority (EFDA). Detailed results and the distribution of failed enalapril maleate samples for the friability test across different sampling sites are shown in Table 4.

The brand Fusix from furosemide tablet samples failed the hardness test and did not meet the specification. These failed drugs were also collected from different sites, as explained in Table 5.

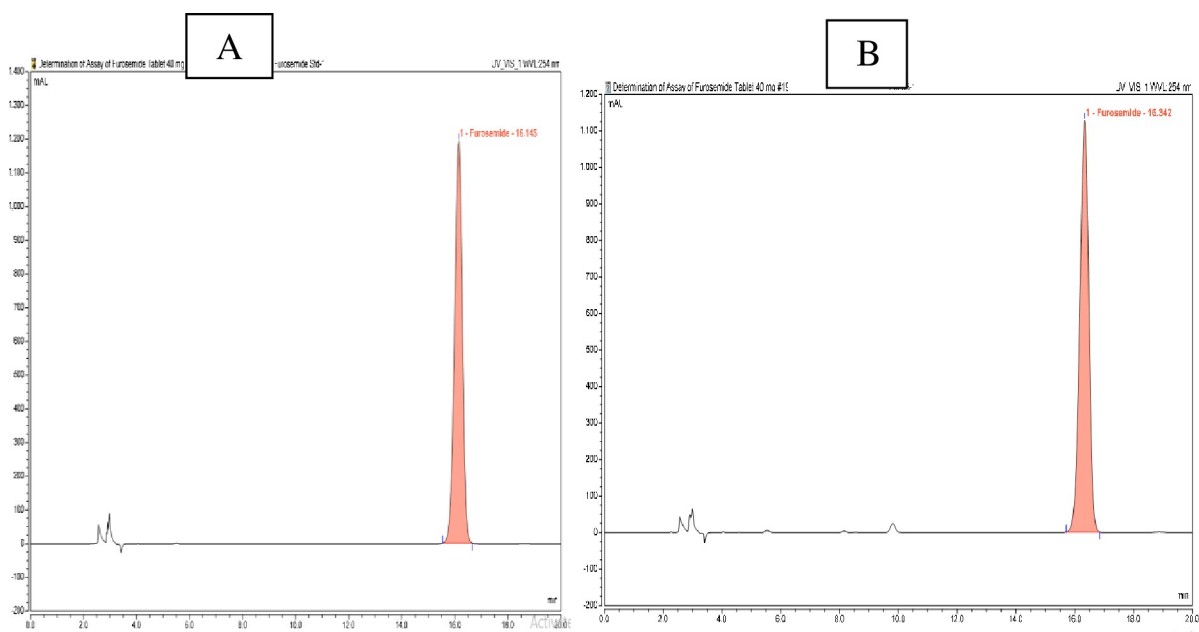

**Fig 6.** HPLC chromatogram results of furosemide standard (A) and sample FM-06 (B).

Different brands of furosemide tablets had also failed the friability test according to the USP specification. The failed samples were also collected from different sites, as shown in Table 6.

The results of friability, hardness, and disintegration tests for all samples are given on S10 and S11 Files.

## Uniformity of dosage units

**Content uniformity of enalapril maleate.** For uniformity of dosage units of enalapril, a content uniformity test was performed using a method specified in USP 2020. According to the specification, the requirements for dosage uniformity are met if the acceptance value of the first 10 dosage units is less than or equal to L1% (Maximum allowed acceptance value) value of 15.

**Table 3. Distribution of failed enalapril samples for hardness test across different sampling sites.**

| S.No | Sample code | Brand name | Batch No. | Collection site | Hardness test value (Mean ± SD) | Conclusion |
|---|---|---|---|---|---|---|
| 1 | EG-02 | Enali-SSP | 06121060030 | Gedeb | 26.9 ± 5.8 | Failed |
| 2 | ED-11 | Enaril-5 | SDJ662 | Dilla | 29.92 ± 4.4 | Failed |
| 3 | EDG-01 | Enali-SSP | 06121060010 | Dilla | 30.7 ± 8.3 | Failed |
| 4 | EGG-01 | Enali-SSP | 06121070030 | Gedeb | 33.3 ± 4.6 | Failed |
| 5 | EYG-01 | Enali-SSP | 06120110010 | Yabelo | 32.8 ± 3.3 | Failed |
| 6 | EY-04 | Enali-SSP | 06121060010 | Dilla | 36.3 ± 2.8 | Failed |
| 7 | EY-06 | Enali-SSP | 06121070020 | Yabelo | 30.2 ± 6.2 | Failed |
| 8 | EY-01 | Enali-SSP | 061201100110 | Yabelo | 35.0 ± 3.3 | Failed |
| 9 | EM-04 | Envas-5 | D21012BX52 | Moyale | 32.8 ± 4.1 | Failed |

SD = Standard Deviation

**Table 4. Distribution of failed enalapril maleate samples for % friability test across different sampling sites.**

| S.No | Sample code | Brand name | Batch No. | Collection site | Friability test value | Conclusion |
|------|-------------|-----------|-----------|-----------------|----------------------|------------|
| 1 | EG-02 | Enali-SSP | 06121060030 | Gedeb | Broken | Failed |
| 2 | EYG-01 | Enali-SSP | 06120110010 | Yabelo | Broken | Failed |
| 3 | EM-08 | Encardil-5 | D00924 | Moyale | Broken | Failed |

*Content uniformity of enalapril maleate using HPLC.* Five replicates of 50 μl enalapril maleate USP reference standard solutions were injected into the HPLC system, and the chromatograms were recorded to evaluate the system suitability parameters, such as the tailing factor (NMT 1.5), theoretical plate number (NLT 300 for enalapril), and % RSD (NMT 2.0). The RSD was 0.99, the theoretical number of plates was 1596.6, and the tailing factor was 0.88. According to these parameters, the system suitability test data for HPLC were set up before the content uniformity of enalapril measurements. It passed the requirements.

This study showed that 8% (2/25) of enalapril maleate samples collected from Moyale had failed for uniformity of dosage units. Brands of enalapril, ACEPRIL and ENCARDIL (with batch numbers 75034 and D00923) from Moyale failed the content uniformity test, with results of 19.42 and 16.3, respectively. The detail results for all assessed samples are shown on S12 and S13 Files.

## Weight variation of furosemide tablets

In this study, even though majority of the samples of furosemide passed, two out of thirty (6.67%) of samples deviated, generating values above and below the pharmacopoeial specification (USP-2020). The failed furosemide of brand FUSIX, with batch numbers 1060513 and 1060373, was collected from Dilla and Yabelo, respectively. All results obtained in regards to this test are shown on S13 File.

## Dissolution

### Dissolution test of enalapril maleate

Five replicates of 50 μL enalapril maleate USP reference standard solutions were injected into the HPLC system, and the chromatograms were recorded to evaluate the system suitability parameters, such as the tailing factor (NMT 1.5), theoretical plate number (NLT 300 for enalapril), and % RSD (NMT 2.0). The system suitability test data for enalapril were fulfilled.

For the analysis of enalapril maleate, a calibration curve was prepared using 6 concentrations within the range of 0.08 mg/mL to 0.28 mg/mL. The regression equation was y = 22.835x + 25.007, where y is the peak response and x is the concentration in mg/mL. With a regression factor of $R^2 = 0.9998$, the calibration curve was found to be linear.

**Table 5. Distribution of failed furosemide samples for hardness test across different sampling sites.**

| S.No | Sample code | Brand name | Batch No. | Collection site | Hardness test value | Conclusion |
|------|-------------|-----------|-----------|-----------------|---------------------|------------|
| 1 | FG-03 | Fusix | 1060453 | Gedeb | 31.4 ± 2.5 | Failed |
| 2 | FYCG-01 | Fusix | 1070013 | Yirgachefe | 13.8 ± 2.6 | Failed |
| 3 | FD-10 | Fusix | 1060513 | Dilla | 19.9 ± 6.5 | Failed |
| 4 | FY-03 | Fusix | 1060423 | Yabelo | 23.7 ± 4.1 | Failed |
| 5 | FG-01 | Fusix | 1060353 | Gedeb | 28.7 ± 3.4 | Failed |
| 6 | FYG-01 | Fusix | 1070103 | Yabelo | 26.9 ± 6.1 | Failed |

**Table 6. Distribution of failed furosemide samples for % friability test across different sampling sites.**

| S.No | Sample code | Brand name | Batch No. | Collection site | % Friability test value | Conclusion |
|------|-------------|------------|-----------|-----------------|-------------------------|------------|
| 1 | FDG-01 | Furosemide | LJ5561 | Dilla | Broken | Failed |
| 2 | FYCG-01 | Fusix | 1070013 | Yirgachefe | Broken | Failed |
| 3 | FD-04 | Furo-Denk | 9ZP | Dilla | 5.268 | Failed |
| 4 | FD-10 | Fusix | 1060513 | Dilla | 1.940 | Failed |
| 5 | FGG-01 | Fusix | 1070063 | Gedeb | Broken | Failed |
| 6 | FD-03 | Fusix | 1070023 | Dilla | 1.022 | Failed |
| 7 | FY-02 | Fusix | 1060373 | Yabelo | Broken | Failed |
| 8 | FM-07 | Fruz | BPL818 | Moyale | Broken | Failed |
| 9 | FM-02 | Furosemide | 2105115 | Moyale | Broken | Failed |
| 10 | FY-03 | Fusix | 1060423 | Yabelo | 1.005 | Failed |

The dissolution profiles of six brands of enalapril maleate passed the USP 43/NF 38 specification limits. As per the specification, enalapril maleate tablets should not release < 80% of the labeled amount within 30 minutes. However, the brand ENARIL-5 with batch number SDJ662, which released an API of less than 80% (77.50 ± 1.91), did not comply with the USP specification. Fig 7(A), depicts the time-dependent dissolution profiles of all brands of enalapril maleate at different sampling points (5, 10, 15, 30, and 60 minutes). Detail exact values are described in the S14 File.

## Dissolution test of furosemide

In the calibration curve, a linear regression equation was y = 0.0616x – 0.0102, where y is the absorbance and x is the concentration in μg/mL. The calibration curve showed a linear relationship between the concentration of the tested samples and the absorbance values over the concentration range of 3.2–11.2 μg/mL ($R^2$ = 0.9994).

This study showed that the mean dissolution test results for different brands of furosemide tablets comply with the USP specification. As per the USP 43/NF 33 specification, furosemide tablets should not release < 80% of the labeled amount within 60 minutes. The mean percentage of API release for collected furosemide samples was at 60 minutes. However, the values obtained ranged from 81.09 ± 0.82 to 88.54 ± 4.12%. However, one brand of Fruz collected in Moyale city with batch number BPL818 failed to meet the USP specification (46.73 ± 0.66% API release). Fig 7(B), depicts the time-dependent dissolution profiles of all brands of furosemide at different sampling points (5, 15, 30, 45, 60, and 75 minutes). The exact values of this test had been included in S15 File.

## Result and calculation for TAMC, TYMC and *E. coli*

**Result and calculations for TAMC.** After incubation, the number of colonies was counted from each Soybean-Casein Digest Agar (SCDA) plate. This number of CFU/gm or CFU/mL was calculated as:

Number of CFU/g or CFU/mL = Average number of colonies on 2 SCDA plates × 10 (Dilution Factor)

Since there were no colonies formed in the study, the samples fulfilled the microbial tests according to the USP <62> criteria [31].

**Result and calculations for TYMC.** After incubation, the numbers of colonies were counted from each Sabourand dextrose agar (SDA) plate. The number of CFU/g or CFU/ml was calculated according to the above formula as in TAMC. The result was as described above

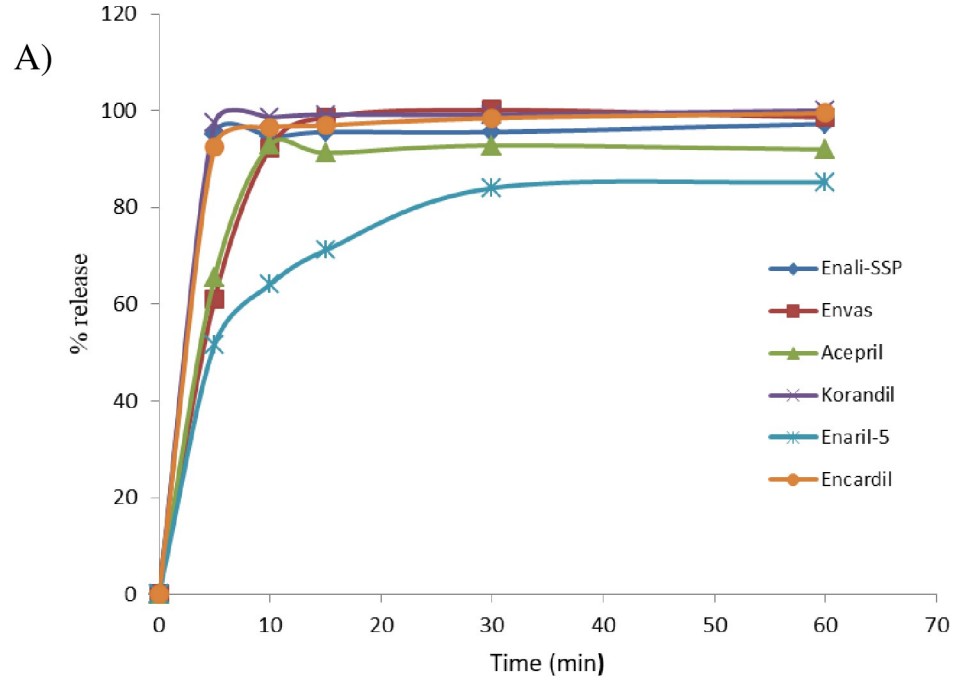

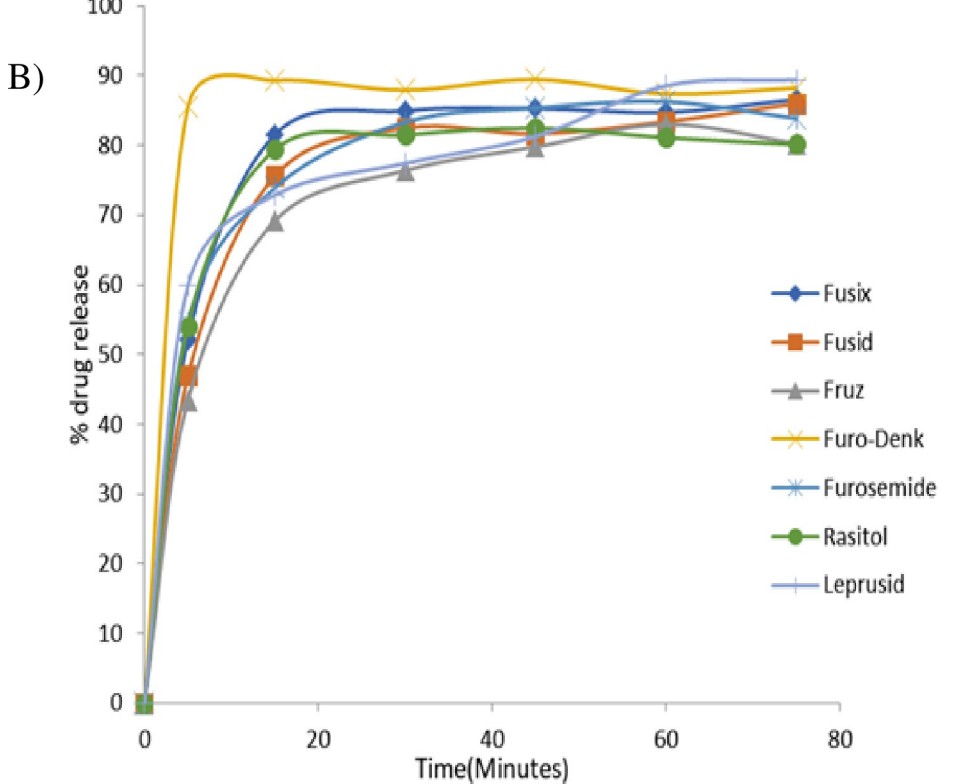

**Fig 7.** Time dependent dissolution profiles of different brand of (A) enalapril, and (B) furosemide tablets.

**Table 7. Overall quality test results of enalapril and furosemide samples.**

| Product (sample) | | | Samples failing quality parameters test | | | | | | |
|---|---|---|---|---|---|---|---|---|---|
| | Visual inspection | Identification | Assay | Uniformity of dosage unit | Hardness | Friability | Disintegration | Dissolution | Overall |
| Enalapril | 12% (3/25) | 0% (not failed) | 0% (not failed) | 8% (2/25) | 36% (9/25) | 12% (3/25) | 0% (not failed) | 4% (1/25) | 48% (12/25) |
| Furosemide | 13.33% (4/30) | 0% (not failed) | 6.67% (2/30) | 6.67% (2/30) | 20% (6/30) | 33.33% (10/30) | 0% (not failed) | 3.33% (1/30) | 53.33% (16/30) |
| Overall | 12.73% (7/55) | 0% (not failed) | 3.64% (2/55) | 7.27% (4/55) | 27.27% (15/55) | 23.64% (13/55) | 0% (not failed) | 3.64% (2/55) | 50.9 (28/55) |

as in TAMC. Hence, the sample passed according to USP <62> specification. The TYMC was considered to be equal to the number of all Colony Forming Units found on the SDA plates; if colonies of bacteria were detected on this medium, they were counted as part of TYMC. Similarly, all negative controls were negative.

**Test for specified microorganisms (*Escherichia coli*).** *Enrichment for Escherichia coli (E. coli)*. The specified microorganism *Escherichia coli* with MacConkey Agar were tested by enrichment. After four days, the observed result was the absence of any colonies on the streaking lines of the petri dishes. Therefore, it was not detected for a defined unit, and thus the sample passed the test. The results for the microbiology tests are included in the supplementary files as S16 File and S2 Fig as well S3 Fig.

## Summary of overall quality assessment results

Of the 55 collected samples, 48% (12/25) of enalapril samples and 53.33% (16/30) of furosemide samples were substandard, failing one or more of the tested parameters. Overall, 50.91% (28/55) of the total collected samples were substandard. Apart from that, all studied samples passed identification and disintegration tests. The details are shown in Table 7.

## Discussion

In this study, the pharmaceutical quality of commonly available brands of enalapril maleate and furosemide tablets in the Gedeo and Borena zones of southern Ethiopia was assessed The results of the visual inspection of the analyzed samples showed problems with packaging. Defects in the physical characteristics of the products were also observed. Poor packaging, the absence of leaflets or package inserts were detected. Furthermore, cracked, eroded, and caked samples were observed during visual inspection. Therefore, these failed samples imply that they were not intact as genuine. However, according to a study performed in Brazil, all samples of inspected enalapril tablets showed satisfactory visual inspection, and none of the analyzed tablets were outside parameters in the visual inspection tools [17]. In contrast to this study, other studies conducted in the Ethiopia capital city Addis Ababa [25] and Bahir Dar city [26] had shown that that all samples of furosemide passed the visual inspection tools.

In the current study, all brands and batches of enalapril maleate and furosemide tablets assessed for identity passed the USP (2020) and BP (2020) methods. As in the present study, all brands of enalapril maleate analyzed passed an identity test in a study performed in Brazil [17] and Guatemala [18]. Another study in Turkey [19] and Nigeria also reported the presence of API in different brands of enalapril maleate. Similarly, a study conducted in Canada [32], Nigeria [23], Libya [7], Ethiopia at Bahir Dar city [26] showed the presence of the furosemide API.

It was revealed that all batches of enalapril maleate met the criteria of the specification in terms of the assay of active ingredients. Similarly, all brands of furosemide performed by USP 2020 passed the specification. However, out of ten furosemide samples analyzed by BP 2020, two samples had less than 95% API content. A similar study conducted in Brazil [17],

Guatemala [18] and Turkey [19] showed that all samples of enalapril maleate passed the USP pharmacopoeial specification. A similar survey conducted by the USP method in two study areas of Ethiopia, Addis Ababa [25] and Bahir Dar city [26], revealed that all samples of furosemide had passed the assay test.

In this study most samples of both enalapril maleate and furosemide samples have fulfilled the acceptance criteria for dosage uniformity. However, 8% (2/25) of enalapril maleate and 6.67% (2/30) of furosemide samples did not meet the pharmacopoeial acceptance criteria (USP, 2020). These findings are not similar to those of previous studies performed in India [16], Brazil [17], Guatemala [18] and southern Nigerian cities [33], in which all the samples passed the dosage uniformity test. The results are not also similar to the studies performed in Canada [32], Nigeria [23], Libya [7] and in Ethiopia Bahir Dar city [26]. This difference may be related to manufacturing practices. The production processes could be the cause of the difference in content uniformity tests. It might be due to variable amounts of excipients and/or APIs use by manufacturers. These variations create particle density difference resulting in different shape and size of tablets. This study revealed that 36% (11/25) and 20% (6/30) of enalapril maleate and furosemide samples failed the hardness tests, respectively. The majority of the failed furosemide brands were Fusix. It is below the minimum value of 40 N [34]. This weak hardness results in a crushing strength that also causes the product to disintegrate and dissolve easily. As a result, those failed samples passed the disintegration and dissolution tests. At the same time, failed brands of enalapril and furosemide in hardness test, they also failed the friability test. This association shows that weak hardness has a direct effect on friability. Similarly, during visual inspection, breakage, chipping, and the caked appearance of the sample might also be due to the low crushing strength of the tablet. Some of these enalapril maleate results are in agreement with the study reported in Turkey, in which all the tablets did not possess enough hardness [19], and southern Nigerian cities [33] that indicated most brands of enalapril maleate failed the hardness test. But, another study conducted in Nigeria that agreed with the current study. Only few of the furosemide brands had failed the hardness test [23]. Additionally, in a study conducted in the Ethiopia capital city Addis Ababa, some brands of furosemide failed to meet the minimum crushing strength [25]. Failure to meet the specification of the hardness test may be due to looseness of inter particulate bonding or the use of low compression pressure by the tablet machine during manufacture [34].

The same three batches of the FUSIX brand failed both the hardness and friability tests simultaneously. Similarly, two batches of ENALI-SSP failed both hardness and friability tests. The samples failing both friability and hardness tests might have common failure factors. This is in agreement with a study in Turkey that some enalapril brands were found to contain failed samples in friability tests that were outside the appropriate limits [19]. A similar study conducted in southern Nigerian cities [33] indicated that most brands of enalapril maleate failed the hardness test. In contrast to these findings, a study conducted in Brazil [17] and the Ethiopia capital city Addis Ababa [25] showed that all tested furosemide samples possess satisfactory results in the friability test.

All brands and batches of enalapril maleate and furosemide tablets had disintegrated in less than 15 minutes. The maximum mean disintegration time results for enalapril maleate and furosemide tablets were 7.33 ± 3.01 and 11.67 ± 2.25 minutes, respectively. This test result agreed with a study conducted in Turkey [19] and southern Nigerian cities [33] in which all brands of enalapril maleate comply with the specification. As in the present study, all brands of furosemide analyzed passed the disintegration test for those studies conducted in Libya [7], Addis Ababa [25] and Bahir Dar, Ethiopia [26].

Dissolution is a test that determines the rate of release of a drug from the dosage form into solution forms across the life cycle [35]. In the present study, the dissolution profile showed

that almost all brands and batches of enalapril maleate and furosemide met the acceptance criteria. However, one batch from each enalapril maleate and furosemide brand failed to release the required amounts as per the pharmacopoeial specification. As in the present study, although two samples failed, almost all brands and batches of enalapril maleate had passed the dissolution test for those studies conducted in Brazil [17], Guatemala [18], and Turkey [19]. A study conducted on furosemide tablets in Brazil [20], Argentina [21], Canada [32], Nigeria [23], and Libya [7] also had shown results that agreed with the current study. However, a similar study conducted on furosemide drugs in two study areas of Ethiopia, Addis Ababa [25], did not agree with the majority of this study while the study in Bahir Dar [26] agreed with this study with respect to the findings of the dissolution test.

The microbial limit test result (TAMC, TYMC, *E. coli*) of furosemide was < 1 CFU/gm (absence of colonies formed), so it had fulfilled the USP <62> specification. It is in agreement with a study conducted in Libya in which all aerobic bacteria and *E. coli* were absent in the five brands of furosemide (< 10 μg/gm of specimen) assessed [7].

## Limitations of the study

The study is performed in certain localized geographic areas; thus, the results might not be applicable to the whole country or in a different context. Nevertheless, the data can be used in situations of similar contexts to fight the worldwide problem of the presence of substandard and falsified (SF) medicines and thereof their public health threat.

## Conclusion

The study attempted to evaluate and compare the quality as well as the physicochemical properties of different brands of enalapril maleate and furosemide tablets. Except for the identification and disintegration test, samples from both furosemide and enalapril failed for pharmacopoeial parameters (assay, friability, dissolution, and uniformity of dosage form) and non-pharmacological parameters (hardness) of the tests performed. For furosemide brands, the microbial limit test in this study revealed that there were no formed colonies of aerobic microbes, yeast, mold and *E. coli*. Generally, the results from this study suggest that approximately half (50.91%) of the enalapril and furosemide products circulating in the markets of the Gedeo and Borena zones at the time of sample collection were substandard (out of specification). Therefore, hypertension medications available on the market may run the risk of losing some of their effectiveness. Hence, it is advisable that the regulatory authority and other stakeholders perform regular actions and monitoring of the medicines available in prone sites such as porous borders to prevent patients from substandard and falsified medicine exposure.

## Supporting information

**S1 Fig. Visual observation defects: Failed visual inspection report of enalapril maleate and furosemide tablets.**
(DOC)

**S2 Fig. Sample conducting the test in the laminar flow hood.**
(DOC)

**S3 Fig. Results of TAMC, TYMC and streaking plate method (Photo).**
(DOC)

**S1 File. Detail information on enalapril maleate samples used for the study.**
(DOC)

**S2 File. Detail information on furosemide samples used for the study.**
(DOC)

**S3 File. General information on number, area and source of collected sample.**
(DOC)

**S4 File. General information on samples used for the study.**
(DOC)

**S5 File. Identification test results of furosemide tablets (USP-2020).**
(DOC)

**S6 File. Identification test results of furosemide tablets (BP-2020).**
(DOC)

**S7 File. Assay value, collection site, brands and batches of enalapril maleate.**
(DOC)

**S8 File. Assay value, brand name and sample site of furosemide tablets (BP 2020).**
(DOC)

**S9 File. Assay value, brand name and sample site of furosemide (USP 43–2020).**
(DOC)

**S10 File. Hardness, friability and disintegration test results of enalapril maleate.**
(DOC)

**S11 File. Hardness, friability and disintegration test results of furosemide tablets.**
(DOC)

**S12 File. Uniformity of dosage units of enalapril maleate by content uniformity.**
(DOC)

**S13 File. Uniformity of dosage units of furosemide by weight variation (n = 3).**
(DOC)

**S14 File. Percentage of enalapril API released at different sampling times (n = 5).**
(DOC)

**S15 File. Percentage of furosemide API released at different sampling times (n = 6).**
(DOC)

**S16 File. Results of TAMC and TYMC on SCDA dishes.**
(DOC)

## Acknowledgments

The authors would like to thank Dilla University and Addis Ababa University School of Pharmacy, Pharmaceutical Chemistry Department for financial support. We keenly thank Humanwell Pharmaceutical Ethiopia PLC for their kind cooperation during medicine quality assessment test studies.

## Author Contributions

**Conceptualization:** Ayenew Ashenef.

**Data curation:** Esayas Tesfaye, Ayenew Ashenef.

**Formal analysis:** Esayas Tesfaye, Tadele Eticha, Ayenew Ashenef.

**Funding acquisition:** Ayenew Ashenef.

**Investigation:** Esayas Tesfaye, Fisseha Woldegiorgis, Ayenew Ashenef.

**Methodology:** Ayenew Ashenef.

**Project administration:** Ayenew Ashenef.

**Resources:** Fisseha Woldegiorgis, Ayenew Ashenef.

**Supervision:** Tadele Eticha, Ayenew Ashenef.

**Writing – original draft:** Esayas Tesfaye, Ayenew Ashenef.

**Writing – review & editing:** Ayenew Ashenef.

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
