## [Decision Letter · Decision Letter 0]

29 Feb 2024

PGPH-D-23-02514

Quality of Medicines for Cardio-Vascular Diseases (CVDs) in the Ethiopian Border with Kenya: The Case of Enalapril Maleate and Furosemide Tablet Quality in Borena and Gedeo Zones.

Dear Dr. Ashenef,

Thank you for submitting your manuscript to PLOS Global Public Health. After careful consideration, we feel that it has merit but does not fully meet PLOS Global Public Health’s publication criteria as it currently stands. Therefore, we invite you to submit a revised version of the manuscript that addresses the points raised during the review process.

This is a well written manuscript  useful in the clinical management of a very common problem. This needs some modifications so that it meet the criteria tfor publication

Since the authors are dealing with a common problem of management of hypertension, you may please also discuss the findings based on the   guidelines for management of hypertension rather than the usual clinical practice

Kindly look into the comments by reviewers regarding the references and grammatical and other mistakes, the correction of which will definitely improve the manuscrip

We look forward to receiving your revised manuscript.

Kind regards,

Suma Krishnasastry, MBBS, MD,DNB

Academic Editor

Journal Requirements:

2. Please provide separate figure files in .tif or .eps format only and remove any figures embedded in your manuscript file. Please also ensure all files are under our size limit of 10MB.

3. Some material included in your submission may be copyrighted. According to PLOS’s copyright policy, authors who use figures or other material (e.g., graphics, clipart, maps) from another author or copyright holder must demonstrate or obtain permission to publish this material under the Creative Commons Attribution 4.0 International (CC BY 4.0) License used by PLOS journals. Please closely review the details of PLOS’s copyright requirements here: PLOS Licenses and Copyright. If you need to request permissions from a copyright holder, you may use PLOS's Copyright Content Permission form.

Potential Copyright Issues:

Fig 2: please (a) provide a direct link to the base layer of the map (i.e., the country or region border shape) and ensure this is also included in the figure legend; and (b) provide a link to the terms of use / license information for the base layer image or shapefile. We cannot publish proprietary or copyrighted maps (e.g. Google Maps, Mapquest) and the terms of use for your map base layer must be compatible with our CC-BY 4.0 license. 

"

4. In the online submission form, you indicated that "The datasets used in this publication are available upon reasonable request from the corresponding author". All PLOS journals now require all data underlying the findings described in their manuscript to be freely available to other researchers, either 1. In a public repository, 2. Within the manuscript itself, or 3. Uploaded as supplementary information.

Additional Editor Comments (if provided):

Reviewers' comments:

Reviewer's Responses to Questions

**Comments to the Author**

1. Does this manuscript meet PLOS Global Public Health’s publication criteria? Is the manuscript technically sound, and do the data support the conclusions? The manuscript must describe methodologically and ethically rigorous research with conclusions that are appropriately drawn based on the data presented.

Reviewer #1: Yes

Reviewer #2: Yes

2. Has the statistical analysis been performed appropriately and rigorously?

Reviewer #1: Yes

Reviewer #2: N/A

3. Have the authors made all data underlying the findings in their manuscript fully available (please refer to the Data Availability Statement at the start of the manuscript PDF file)?

Reviewer #1: Yes

Reviewer #2: Yes

4. Is the manuscript presented in an intelligible fashion and written in standard English?

Reviewer #1: Yes

Reviewer #2: Yes

5. Review Comments to the Author

Reviewer #1: The study conducted addresses the scope and objectives of the study. The results section could have been summarized briefly as the tables and figures are self explanatory. Since the results section is quite elaborate, the discussion section has a lot of redundancy with the results statements.

Reviewer #2: The manuscript is well written manuscript with some minor issues to be addressed before published.

1. Some of the references cited in this manuscript are dated very old and need to be replaced with the latest ones.

2. Its been mentioned in the manuscript that the enalapril and the furosemide are the widely prescribed medicines for hypertension. No guidelines for hypertension made such a recommendation. The information provided in the manuscript need to be correct and accurate. Furosemide is a loop diuretic commonly used to manage conditions like edema and congestive heart failure. While it can lower blood pressure, it is not a first-line antihypertensive agent.

Loop diuretics, including furosemide, are not recommended as the primary treatment for hypertension due to the lack of robust outcome data.

3. The grammatical mistakes found throughout the manuscript decreases the weightage of the manuscript.

6. PLOS authors have the option to publish the peer review history of their article (what does this mean?). If published, this will include your full peer review and any attached files.

**Do you want your identity to be public for this peer review?** For information about this choice, including consent withdrawal, please see our Privacy Policy.

Reviewer #1: **Yes: **Abdul Nazer Ali

Reviewer #2: No

---

## [Editor Report · Decision Letter 1]

19 Apr 2024

Quality of Medicines for Cardio-Vascular Diseases (CVDs) in the Ethiopian Border with Kenya: The Case of Enalapril Maleate and Furosemide Tablet Quality in Borena and Gedeo Zones.

PGPH-D-23-02514R1

Dear Asst. Professor Ashenef,

We are pleased to inform you that your manuscript 'Quality of Medicines for Cardio-Vascular Diseases (CVDs) in the Ethiopian Border with Kenya: The Case of Enalapril Maleate and Furosemide Tablet Quality in Borena and Gedeo Zones.' has been provisionally accepted for publication in PLOS Global Public Health.

Best regards,

Suma Krishnasastry, MBBS, MD,DNB

Academic Editor